# A predicted structural interactome reveals binding interference from intrinsically disordered regions

Junhui Peng ⓘ*, Li Zhao ⓘ*

Laboratory of Evolutionary Genetics and Genomics, The Rockefeller University, New York, New York, United States of America

* jpeng@rockefeller.edu (JP); lzhao@rockefeller.edu (LZ)

## Abstract

Proteins function through dynamic interactions with other proteins in cells, forming complex networks fundamental to cellular processes. While high-resolution and high-throughput methods have significantly advanced our understanding of how proteins interact with each other, the molecular details of many important protein-protein interactions are still poorly characterized, especially in non-mammalian species, including *Drosophila*. Recent advancements in deep learning techniques have enabled the prediction of molecular details in various cellular pathways at the network level. In this study, we used AlphaFold2 Multimer to examine and predict protein-protein interactions from both physical and functional datasets in *Drosophila*. We found that functional associations contribute significantly to high-confidence predictions. Through detailed structural analysis, we also found the importance of intrinsically disordered regions in the predicted high-confidence interactions. Our study highlights the importance of disordered regions in protein-protein interactions and demonstrates the importance of incorporating functional interactions in predicting physical interactions between proteins. We further compiled an interactive web interface to present these predictions, facilitating functional exploration, comparative analysis, and the generation of mechanistic hypotheses for future studies.

## Author summary

Understanding which proteins interact with each other and how they interact is essential for uncovering fundamental biology and for identifying new pathways involved in health and disease. However, identifying protein-protein interactions experimentally is often challenging and error-prone, and many organisms still lack comprehensive interaction maps. In this study, we use AlphaFold2 Multimer, a powerful artificial intelligence tool, to generate high-confidence predictions of PPIs in *Drosophila melanogaster*, a widely used model organism. We

**Data availability statement:** Data availability All predicted structures are available in figshare (doi.org/10.6084/m9.figshare.29319614) and the DmelPPI website (https://melppi.github.io). Code availability All scripts and codes are available in GitHub (https://github.com/LiZhaoLab/DmelPPI and https://github.com/melppi/melppi.github.io).

**Funding:** This work was supported by National Institutes of Health (NIH) MIRA R35GM133780 and the Allen Distinguished Investigator Award from Paul G. Allen Family Foundation to L.Z. The work was also partly supported by the Shapiro-Silverberg Fund for the Advancement of Translational Research to J.P. The funders had no role in study design, data collection and analysis, decision to publish, or preparation of the manuscript.

**Competing interests:** The authors have declared that no competing interests exist.

highlight the importance of incorporating often neglected functional associations when predicting protein-protein interactions at a genomic scale. The predictions enable us to examine how intrinsically disordered regions can mediate binding across large interaction networks, revealing widespread, structurally plausible interactions *in vivo*. Overall, our work demonstrates how AlphaFold predictions can greatly expand our understanding of the structural forces that shape protein interaction networks and help reveal hidden layers of cellular complexes and pathways.

## Introduction

Cells are dynamic environments where thousands of proteins interact simultaneously at any given time, and these protein-protein interactions (PPIs) are essential for various biological functions. PPIs can range from stable, permanent associations – such as those within stable protein complexes and molecular machinery – to transient, short-lived interactions with a wide range of affinities and lifetimes, as seen in signal transduction pathways [1–4]. Understanding the organization and dynamics of the cellular PPI network is crucial for a comprehensive picture of cellular functions. Over the past decades, diverse high-throughput methods have been developed and used to study these interactions in different organisms, presenting a systematic overview of how proteins work together in cells [5]. For instance, affinity purification mass spectrometry [6] has been instrumental in identifying protein complexes. Yeast two-hybrid systems [7] and other in vitro techniques such as cross-linking mass spectrometry [8] have also been used to map PPIs at high throughput. More recently, in vivo proximity labeling methods such as BioID [9] and TurboID [10] have emerged as powerful tools for capturing dynamic protein interactions within living cells.

*Drosophila* is an important model species for biological and biomedical research. Despite significant efforts to study *Drosophila* protein-protein interactions both in vivo and in vitro, many uncertainties remain in existing PPI databases, such as false positives [11], inconsistency between databases [12], and underrepresentation of transient or weak interactions [5]. The combination of complex tissues and small body sizes poses unique challenges in biochemistry compared to other model organisms such as rodents, particularly due to the difficulty in obtaining sufficient amounts of protein samples from specific tissues for high-quality experiments such as affinity purification mass spectrometry. Additionally, methods such as yeast two-hybrid may introduce errors due to system differences [13]. Therefore, it is essential to evaluate these databases to gain a deeper understanding of the reliability and biases of previous methods [11,14]. Furthermore, while much is known about the interactions of ordered structural components, characterizing the interaction details involving disordered proteins or disordered regions remains challenging [1,15–21]. It has been shown that disordered regions can adopt diverse binding modes, ranging from transient fuzzy complexes to disorder-to-order transitions upon binding [15–17,19–22]. Recent experimental and computational studies further highlight conditionally folding

as an important disorder-to-order binding mechanism [19,23]. Despite these studies, systematic characterization of different disorder binding modes and their contributions to protein-protein interactions at the proteome scale remains limited.

AlphaFold2 Multimer [24,25] has provided a new tool for evaluating direct and indirect protein interactions, potentially distinguishing true, stable interactions from experimental artifacts [18,26–28], and enhancing our understanding of the structure and function of interaction networks. From an evolutionary biological perspective, AlphaFold2 Multimer is also a powerful tool for studying the interactions and changes in interactions of homologous proteins across different lineages [29].

In this study, we combine AlphaFold2 Multimer and the STRING database [14] to examine and predict the structural details of protein-protein interactions in *D. melanogaster*. We aim to systematically and comprehensively evaluate different sources of protein-protein interactions within the *Drosophila* genome, including physical interactions inferred from experiments or homology and functional associations derived from co-expression, co-existence in the same pathways, text-mining, genomic context, etc. Our study provides a resource for the structural details of possible physical interactions in the *Drosophila* genome, and highlights the importance of incorporating functional PPIs in future high-throughput computational studies. We also reveal the important and prevalent roles of intrinsically disordered regions in the *Drosophila* PPI network. We further compiled our analysis and results into an interactive web interface for users to explore, validate, and expand protein interaction hypotheses.

## Results

### Predicted high-confidence *D. melanogaster* physical protein-protein interaction network

We downloaded all the protein-protein interactions (PPIs) with the highest STRING confidence scores (≥0.9) from the STRING database, version 11.5 [14], which include both direct physical interactions and indirect functional associations. The STRING database provides five main, non-exclusive categories of supporting evidence for the PPIs, including experimental, co-expression, curated databases, genomic context, and text-mining of published literature. In addition to these five categories, the STRING database has also provided a physical subnetwork by incorporating evidence from the curated database organized in IMEx [30] and BioGRID database [31]. Since it has been indicated that functionally associated proteins, such as co-expressed proteins, are more likely to interact with each other than random protein pairs [32–34], we did not restrict our analysis to the physical subnetwork, as our goal is to systematically predict and examine all possible protein-protein interactions.

We used AlphaFold2 Multimer to predict the structures of 27,711 protein pairs from the STRING database with the highest STRING confidence scores (≥ 0.9), involving approximately 5000 protein-coding genes clustered at 50% protein-sequence identity (Methods). The 27,711 protein pairs can be divided into five mutually exclusive categories according to STRING confidence scores in the five different supporting evidence categories, including 7838 in experimental, 7088 in co-expression, 6764 in curated databases, 5995 in text-mining, and 6 in genomic context (S1 Fig). We used pDockQ to assess AlphaFold2 Multimer prediction confidence [26,35]. Among the 27,711 protein pairs, 8,101 pairs (29.2%) were predicted to have direct physical interactions at acceptable confidence with pDockQ ≥ 0.23, and 3,621 pairs (13.1%) at high confidence with pDockQ ≥ 0.5. A summary of the predictions can be found in S1 Appendix. This suggests that a relatively small percentage of the previously reported functional associations may have direct physical interactions.

We further examined whether the 3,621 predicted high-confidence PPIs (pDockQ ≥ 0.5) are supported by the STRING-curated physical subnetwork. Among these predicted high-confidence PPIs, 52% (1867 pairs) are supported by highest STRING confidence scores for physical interaction (≥ 0.9, S2 Fig), ~70% (2541 pairs, which include the 1867 pairs with scores ≥ 0.9) by high STRING confidence scores (≥ 0.7, S2 Fig), and ~81% (2929, which include the 2541 pairs with scores ≥ 0.7) by median STRING confidence scores (≥ 0.4, S2 Fig). When further categorizing the 3,621 predicted high-confidence PPIs into five STRING database categories according to their highest STRING confidence scores, we

found that 31.6% (1144) are best supported by experimental data, 13.7% (496) by co-expression, 19.7% (715) by curated database, 34.9% (1263) by text-mining, and 0.1% (3) by genomic context, specifically gene fusions. These results suggest the robustness of our AlphaFold2 Multimer predictions, with ~81% of the predicted high-confidence PPIs supported by reasonable physical interaction evidence in the STRING database. Interestingly, many of the remaining ~19% predicted interactions that lack physical evidence in STRING are supported by indirect functional associations, such as previously reported co-expression and co-occurrence in cellular pathways, highlighting the importance of incorporating functional associations in predicting PPI structures.

## AlphaFold2 Multimer predicts core physical interaction networks

We combined network clustering and gene ontology (GO) enrichment analysis to investigate the overall picture of the highly connected network formed by the 3,621 high-confidence physical PPIs (Fig 1). Overall, the 3,621 predicted high-confidence PPIs involved 2,740 proteins, with an average of 2.6 interactions per protein. We found that the PPI network was composed of one large, connected subnetwork comprising 2,630 PPIs, 11 relatively small, connected subnetworks comprising 10–43 PPIs, 128 very small, connected subnetworks comprising 3–9 PPIs, and 228 singleton PPIs (S2 Appendix). The largest connected subnetwork with 2,630 PPIs involves 1,534 proteins, while the second largest subnetwork with 43 PPIs includes 32 proteins (Fig 1b). The other subnetworks are relatively small, each containing fewer than

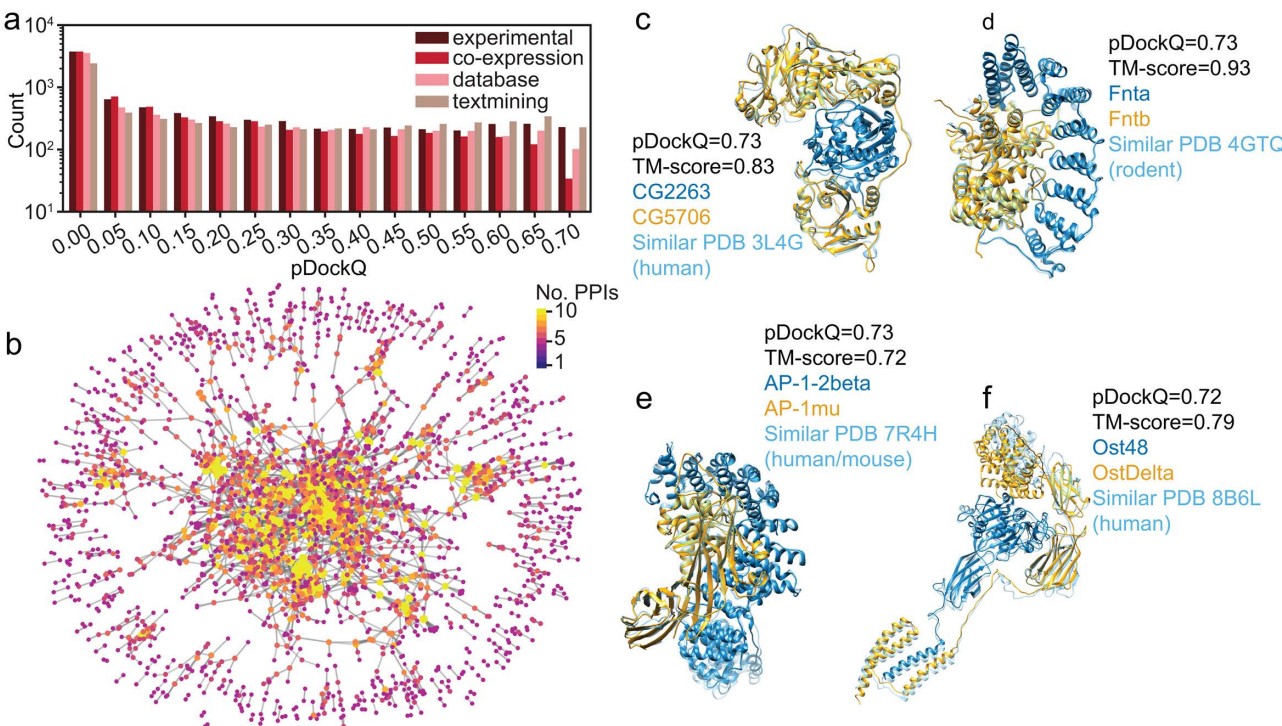

**Fig 1. Overall pictures of AlphaFold2 Multimer predictions of *D. melanogaster* PPIs in the STRING database. a)** The number of predicted high-confidence PPIs supported by the highest evidence scores in the STRING database. The other 6 PPIs inferred from genomic context were not shown because of small sample sizes. **b)** The PPI network of predicted high-confidence PPIs by AlphaFold2 Multimer. Each node represents a protein, and edges denote predicted high-confidence interactions. Each node was colored according to the number of its partners. The dense and overlapping layout reflects the highly interconnected nature of the interactome. **c), d), e),** and **f)** Predicted PPI structures of CG2263/CG5706, Fnta/Fntb, AP-1-2beta/AP-1mu, and Ost48/OstDelta, respectively. The predicted PPIs were shown in blue and orange, each representing one protein. Closely matched PDB structures were shown in cyan. The source organisms of the PDB structures were also labeled.

30 proteins. The 228 isolated PPIs correspond to 456 proteins. Functional enrichment analysis of the largest connected subnetwork showed highly significant enrichment in biological processes such as transcription, translation, gene expression, and metabolism of macromolecules (S1 Table), highlighting the interconnected nature of these fundamental cellular processes. In the second largest connected subnetwork, PPIs were mostly enriched in reproduction (S2 Table), and the third largest was mostly enriched in signaling-related biological processes (S3 Table). We further removed ribosomal proteins from the high-confidence PPI network, as ribosomal proteins were reported to be positively charged and bind to negatively charged proteins non-specifically. The resulting network has similar connectivity with 2585 nodes, 3290 interactions, and an average of 2.5 interactions per protein (S3 Fig). The results further support the interconnected nature of the high-confidence PPIs.

We further applied Markov Clustering (MCL) Algorithm [36] to identify clusters of interacting proteins that may represent core subunits of protein complexes or core components in signaling pathways. The results showed that the top clusters represent components of major cellular complexes, such as ribosomes, the SNARE complex, the transcription initiation complex, spliceosome, the mitochondrial respiratory complexes, the mediator complex, the ubiquitin ligase complex, the respiratory chain complex, the proteasome complex, and part of major signaling pathways, such as the CDK-dependent transcription regulation, SUMO pathway, GPCR signaling pathway, TORC2 signaling pathway, etc (S4 Fig, S3 Appendix). This finding aligns with the results of the above GO functional enrichment analysis.

To further illustrate the predicted PPI structures, we showcase some examples where interactions were supported by various evidence types in the STRING database (Fig 2). For instance, the CG2263/CG5706 and Fnta/Fntb were physical interactions supported by high-throughput affinity purification mass spectrometry in *Drosophila* [37]. Two other examples, AP-1–2beta/AP-1mu and Ost48/OstDelta were primarily functional associations derived from text-mining either in *D. melanogaster* (AP-1–2beta/AP-1mu) or transferred by homology from other organisms (Ost48/OstDelta). In all cases, the predicted PPI structures closely match homologous complexes in other organisms, with TM-scores of 0.83, 0.93, 0.72, and 0.79, respectively. Notably, in the last example of Ost48/OstDelta – where only exploratory physical interaction evidence is available in *D. melanogaster* – STRING database version 11.5 assigned a high confidence physical interaction score based on homology transfer and AlphaFold2 Multimer successfully predicted their interactions. This highlights that even in the absence of direct interaction evidence in *D. melanogaster*, high STRING confidence scores can be informative.

## Predicted high-confidence interactions with limited or no support in the STRING physical subnetwork

To gain an understanding of how indirect, functional associations – such as co-expression – contribute to the predicted high-confidence PPI network, we examined the subset protein pairs with limited or no support in the STRING physical subnetwork (STRING confidence score < 0.4). The analysis identified 692 predicted high-confidence PPIs, involving 918 proteins (Fig 2a, S4 Appendix). Among them, we observed a large, connected subnetwork consisting of 136 proteins and 105 PPIs as well as three median-sized connected subnetworks with 10–50 PPIs each (Fig 2a and 2b). The remaining PPIs formed 82 small, connected subnetworks with 3–9 PPIs and 176 singleton PPIs.

To examine the biological relevance of these predicted PPIs, we manually examined several interactions. We identified potential PPIs that were either characterized after the release of the STRING database version 11.5 [14] or suggested by genetic experiments but lacked evidence of physical interaction. For example, Ran/Nup50 and Ptp61F/Stat92E were both scored as high-confidence functional associations but low-confidence physical interactions in the STRING database. However, recent high-resolution structures of the interactions between the orthologs of Ran-Nup50 in humans suggest that they directly interact [38]. Our AlphaFold2 Multimer analysis supports a high-confidence interaction of Ran/Nup50 and Ptp61F/Stat92E (Fig 2c-2e). Notably, the structure of Ran/Nup50 (PDB code 7MO0) was released after the AlphaFold2 Multimer training cutoff date (September 30, 2021) and thus was not included in model training [24]. In the other example, Stat92E is a transcription factor in the JAK/STAT signaling pathway, and Ptp61F is a protein tyrosine phosphatase that regulates the pathway. There is no direct evidence that Ptp61F directly interacts with Stat92E in *Drosophila*. However,

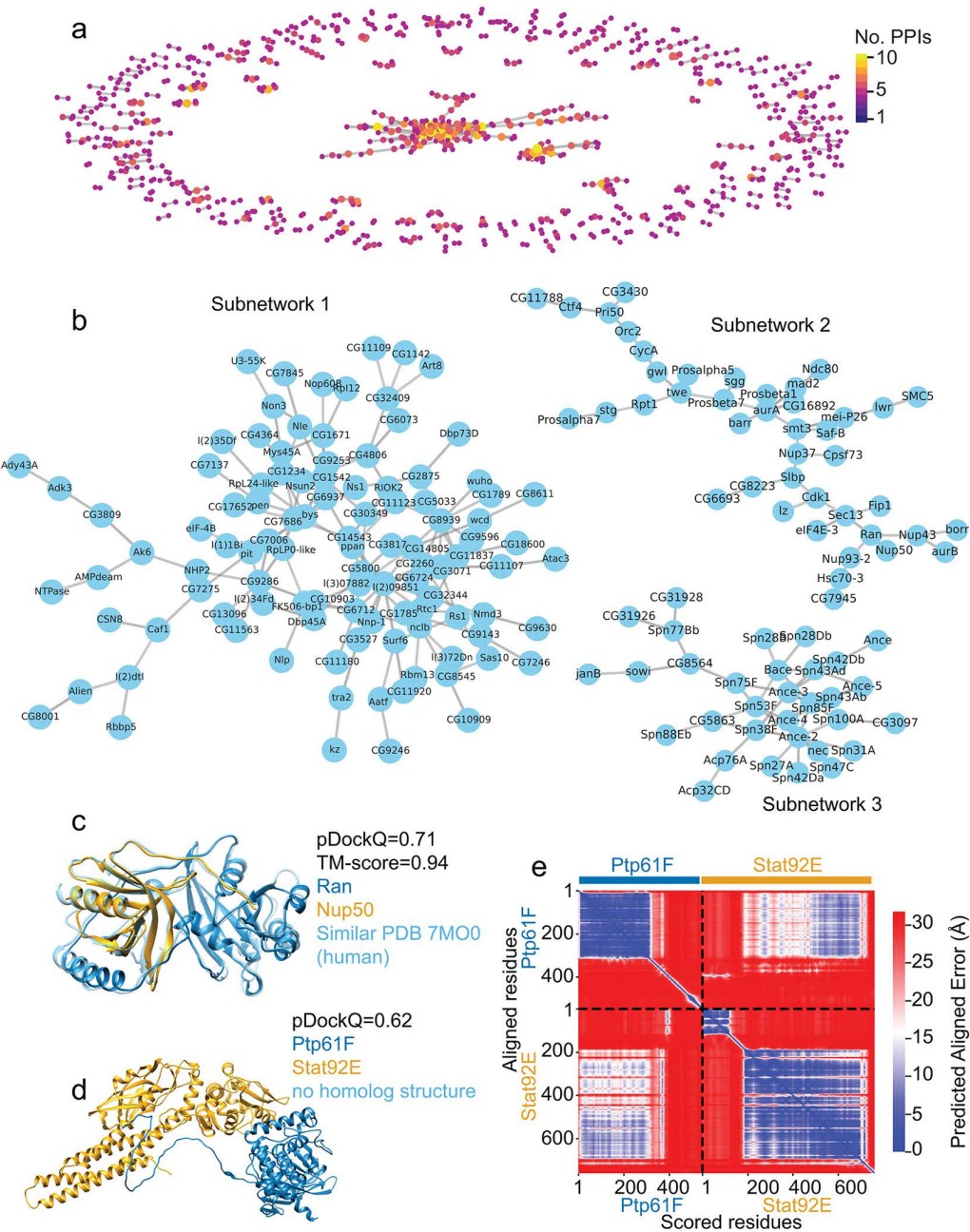

**Fig 2. Predicted high-confidence interactions with limited or no support in the STRING physical subnetwork. a)** The PPI network of predicted high-confidence interactions that have STRING confidence score < 0.4 in the STRING physical subnetwork. **b)** The two largest connected subnetworks from a). **c)** The predicted interactions between Ran (blue) and Nup50 (orange) closely match the orthologous complex (cyan, PDB code 7MO0) in humans. **d)**. The predicted interactions between Ptp61F (blue) and Stat92E (orange). Flexible regions that are not in the interfaces were not shown in (c) and (d). The predicted full-length model of Ptp61F/Stat92E with flexible regions is shown in S6 Fig. **e)** The predicted aligned error map of Ptp61F-Stat92E interactions. The off-diagonal blue regions indicate that Ptp61F residues 1 to 379 might interact with Stat92E residues 186 to 697 with relatively small predicted aligned errors.

indirect evidence indicates that knockout of Ptp61F in *Drosophila* increases the level of phosphorylated Stat92E [39,40], suggesting that Ptp61F potentially interacts with phosphorylated Stat92E to mediate its dephosphorylation. Overall, high-confidence PPI predictions will provide valuable hypotheses for future functional and mechanistic studies.

## Most predicted dimers involve binding through disordered regions

To further understand the interaction details of these PPIs at the genomic scale, we analyzed the interaction interfaces of each PPI and classified them into different binding modes, including binding through ordered regions and IDRs (Methods). The latter were further classified into three modes, which are disordered binding, coil-to-order binding, and conditionally folded (CF) binding (Methods). CF binding was hypothesized to occur predominantly via conformational selection [23,41], while coil-to-order binding is more likely driven by a combined mechanism of conformational selection and induced fit, where induced fit represents an entropy-favored mechanism in which a disordered region folds into a specific structure upon binding, adapting its shape to fit the interacting partner [22]. Our analysis shows that while almost all the predicted high confidence PPIs involve ordered regions (98.5%, 3566/3621), a comparable proportion of PPIs (98.8%, 3580/3621) involve IDRs, either entirely or in part (Fig 3a). PPIs that involve long IDRs (≥30 residues in length) also represent a notable fraction (75.1%, 2719/3621) (Fig 3a). When classifying each PPI that involves IDR to the above mentioned three categories, we found many interactions involving conditionally folded binding (51.7%, 1873/3621) and coil-to-order binding (21.0%, 762/3621) (Fig 2a). The results highlight the important role of IDRs in mediating protein-protein interactions at the genomic scale.

Note that the three binding modes are not mutually exclusive when two proteins interact, as there might be multiple IDRs in the complex that belong to different binding modes. For example, Ku80 and Irbp are important components in DNA damage response. The predicted PPI structure was highly confident with a pDockQ of 0.73, which matches the homologous human complex structure (PDB code 7Z6O, Fig 3b). Through detailed structural analysis, we found several disordered regions in both Ku80 and Irbp monomers involved in their interactions (Fig 3c). Among these disordered interfaces, residues 246–272 in Ku80 (Ku80_246–272) undergo coil-to-order transitions, where the coil secondary structure in Ku80 monomer becomes a strand in the complex (Fig 3d, left panel). Interestingly, the Irbp interface (residues 276–302) that interacts with Ku80 (residues 246–272) also undergoes a coil-to-order transition (Fig 3d, bottom panel). Besides these interfaces, we also observed that some interfaces between Ku80 and Irbp are conditionally folded regions. For example, two of the interfaces in Ku80, Ku80_423–429 and Ku80_509–534, were predicted to be disordered (Fig 3c, left panel). However, they were also predicted to have high pLDDTs (≥ 70) in the Ku80 monomer and contain helices in the monomer (Fig 3c, left panel). Such structural versatility may enable proteins such as Ku80 and Irbp to fine-tune their interactions in response to cellular conditions, particularly during processes such as DNA damage response. The complex roles and various binding modes of IDRs suggest that a single PPI can involve multiple binding mechanisms simultaneously, highlighting the structural complexity and dynamic nature of disordered regions in mediating protein-protein interactions.

## Example of IDR interfaces with different binding mechanisms and evolutionary origins

IDR interfaces can either be conserved or lineage specific. Here, we present examples in which the IDR interfaces have different evolutionary origins and undergo different binding mechanisms. One example is the predicted PPI structure between Sec61alpha and CG13426 (Fig 4a, left panel), which resembles the human homolog structure (PDB code 1LDJ). Sec61alpha and CG13426 are both subunits of the translocon; their interactions are important for the transportation of peptides onto the ER membrane. The disordered interface in Sec61alpha is residues 47–55, which does not involve coil-to-order transitions upon binding. The interface in CG13426, residues 46–104, is predicted to be conditionally folded, i.e., it is predicted to be disordered but with high pLDDT (>70) in the monomer state (Fig 4A, right panel). Despite these differences in binding mechanism and intrinsic disorder propensity, both interface regions are highly conserved across diverse organisms (Fig 4b).

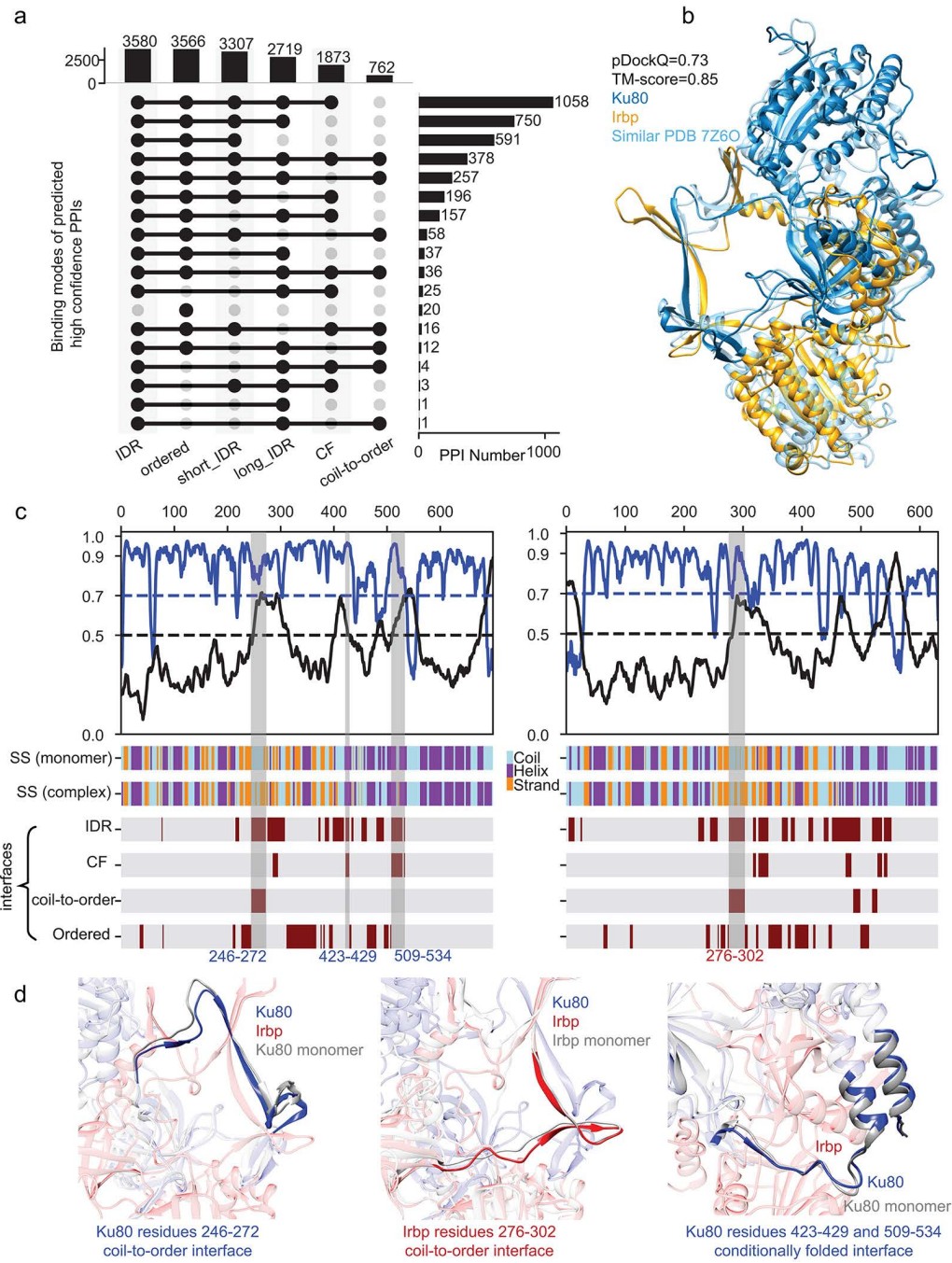

**Fig 3. Many PPIs involve IDRs. a)** Number of PPIs with disordered (IDR), ordered, short IDR, long IDR, conditionally folded (CF), and coil-to-order interfaces. **b)** An example of the predicted PPI structure of Ku80/Irbp with its similar PPI structure of their homologs in humans (PDB code 7Z6O). **c).** pLDDT (blue line) and intrinsic structural disorder (ISD, black line) of Ku80 (left column) and Irbp (right column) in their monomer states. pLDDT values were rescaled to 0 to 1 by dividing by 100. The secondary structures of Ku80 and Irbp in monomers, SS (monomer), and complex states, SS (complex), were shown in the middle panels, with coil in lightblue, helix in purple, and strand in orange. Different interfaces were shown in the bottom panels, including disordered binding interface (labeled as IDR), conditionally folded binding interface (labeled as CF), coil-to-order binding interface (labeled as coil-to-order), and ordered binding interface (labeled as ordered). The presence of one binding mode at a specific residue index is indicated by darker maroon. **d)** Examples of disordered (left panel), coil-to-order (middle panel), and CF (right panel) interfaces in the predicted Ku80/Irbp structure. The corresponding regions in Ku80 and Irbp were shaded in **(c)**.

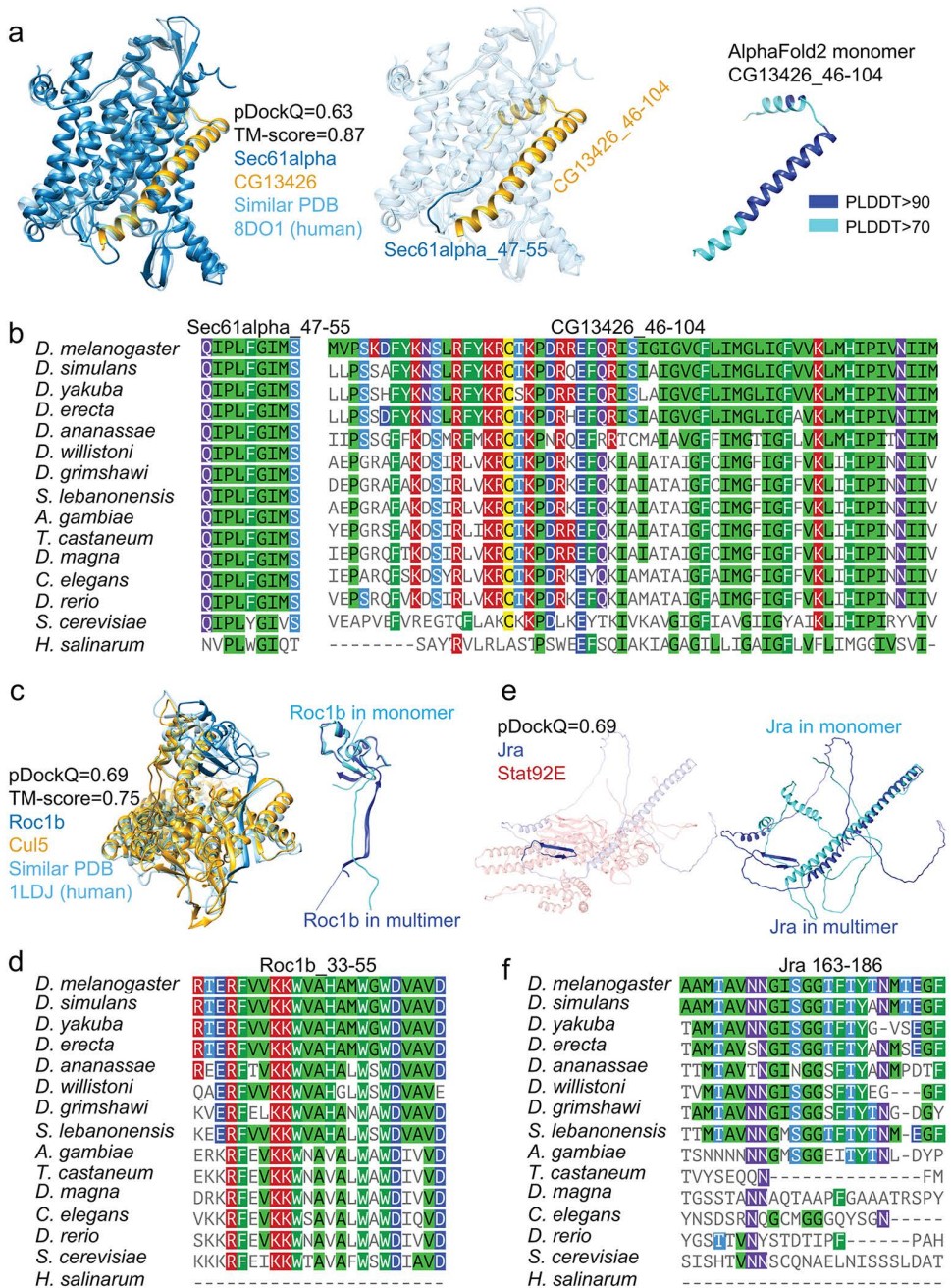

**Fig 4. PPI examples illustrate different IDR interfaces with different evolutionary origins.** a) interact through a coil region in Sec61alpha and a conditionally folded region in CG13426. **b)** Both interfaces of Sec61alpha and CG13426 are highly conserved and have very early evolutionary origins in eukaryotes. **c)** Roc1b undergoes coil-to-order transitions when binding to Cul5. **d)** The coil-to-order interface for Roc1b is highly conserved and has early evolutionary origins. **e)** Jra undergoes coil-to-order transitions when binding to Stat92E. **f)** The coil-to-order interface for Jra has a relatively recent origin and is highly conserved after its origin.

In another example, Roc1b/Cul5 is a key interaction in the multisubunit Cullin-RING E3 ubiquitin ligase. The predicted PPI structure showed that the interface between Roc1b and Cul5 involves a long β strand ([Fig 4c](), left panel). However, this strand was not observed in the monomer structure of Roc1b ([Fig 4c](), right panel). Thus, we categorized this interface

as a coil-to-order interface. We showed that this interface has an early evolutionary origin and is highly conserved across different taxa (Fig 4d, left panel). A third example is a recently originated coil-to-order interface in Jra (Fig 4e and 4f). In this example, Jra residues 163–186 were disordered, forming a coil structure in the monomer state (Fig 4e). It forms an antiparallel β strand structure upon binding to its partner Stat92E. The interface is relatively new in evolution, which can be dated back to arthropods (Fig 4f).

**An interactive web interface for exploring, validating, and expanding protein interaction hypotheses**

Building on our predictions and analysis, we developed an interactive web platform for scientists to explore and generate new hypotheses from all the 27,711 predicted interactions. The platform is accessible at https://melppi.github.io. The homepage of the platform provides a summary of the full prediction datasets, including FBpp identifiers, gene symbols, model quality metrics, and a quick summary of prediction confidence. A search function allows users to query genes or proteins of interest by gene symbols and FBpp identifiers. Users can search for a protein pair using FBpp identifiers, gene symbols, or FBgn numbers, separated by a space, e.g., "Ku80 Irbp". After selecting an interaction, users are directed to a detailed result page by clicking "View Details". In the details page, users can view the predicted structures, predicted aligned error map, and explore the interaction details, such as different binding modes of the interfaces. The interaction details were only shown for predictions with high and acceptable confidence at pDockQ ≥ 0.23 and other possible interactions with ipTM ≥ 50 and PAE ≤ 10 Å. The structure view is interactive, allowing users to select and highlight interface residues with a single click. Besides, the detailed result page also provides protein annotations, the position of the protein pair in the predicted high-confidence PPI network, a link to the interaction evidence in the STRING database, etc (Fig 5). The website is intended not only to present the predictions, but also to facilitate functional exploration, comparative analysis, and the generation of mechanistic hypotheses. By integrating structural and functional information, it offers a useful starting point for both experimental follow-up and computational investigation.

## Discussion

In this work, we used AlphaFold2 Multimer to examine and predict protein-protein interactions of *D. melanogaster* proteins in the STRING dataset. Notably, we found that 81% of AlphaFold's high-confidence predictions aligned with some physical evidence in STRING (score ≥0.4, with about half ≥0.9). We also showed that the remaining 19% predictions, while lacking reasonable direct evidence of physical interaction, may still represent biologically relevant interactions. Our predictions highlight the importance of incorporating functional associations in guiding the prediction of protein-protein interactions. In the meantime, our predictions offer a high-confidence resource for hypothesis generation in future functional and mechanistic studies.

Our study also reveals notable discrepancies between computationally or functionally inferred protein-protein interactions and direct physical interactions. Among the 27,711 high-confidence STRING pairs (score ≥ 0.9) analyzed, only 13.1% (3,621 pairs) were predicted as high-confidence physical interactions by AlphaFold2 Multimer (pDockQ ≥ 0.5). Several factors may account for this discrepancy: (1) some interactions are merely functional associations and do not physically interact – for example, proteins that exhibit genetic interactions or co-occur in cellular pathways but do not physically interact with each other; (2) previously identified interactions from high-throughput studies may include artifacts – for example, indirect interactions that require other components, or those involving low-abundance proteins that are not reliably detected [42]; and (3) AlphaFold2 Multimer combined with pDockQ overlooked predicted aligned error and may preferentially predict interactions with large interfaces [43], potentially overlooking biologically meaningful but transient, real interactions [44]. Indeed, except for the total 8,108 acceptable interactions (including 3,621 high-confidence ones), there are another 2,910 interactions with minimal inter-protein predicted aligned error smaller than 10 Å (PAE ≤ 10 Å) or interface pTM greater than 50 (ipTM ≥ 50), indicating these interactions may be potentially biologically relevant but could be dynamic and transient [18].

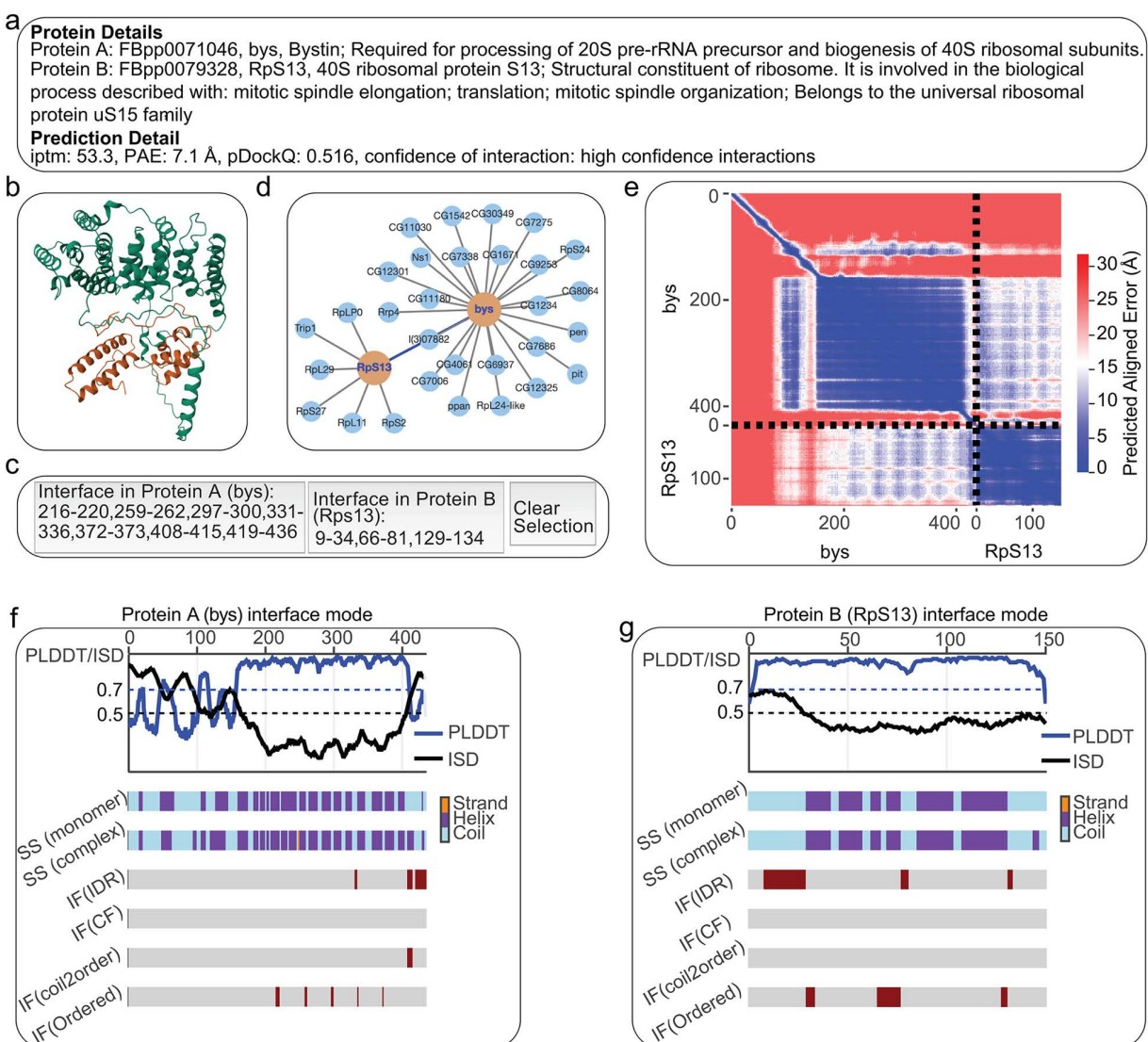

**Fig 5. Information in the detailed page showing interactions between bys and RpS13 in *D. melanogaster*.** The detailed page includes protein annotations **(a)**, prediction details **(a)**, interactive structure viewer (b) with buttons to select interface residues **(c)**, position in predicted high confidence PPI network **(d)**, predicted aligned error heatmap **(e)**, and interface binding modes for each of the proteins in the complex **(f and g)**.

One notable observation was that functional associations that show the highest STRING confidence scores in co-expression and text-mining, had similar proportions predicted as high confidence physical interactions as those supported primarily by experimental evidence (Fig 1a). This highlights the broader need for experimental validation of such associations. In particular, AlphaFold2 Multimer predictions supported by functional evidence such as text mining and lacking evidence from STRING physical subnetwork warrant further experimental investigation. Independent high-throughput experimental approaches such as cross-linking mass spectrometry or targeted mutagenesis could help resolve ambiguities in these cases, including whether the predicted interaction is direct or indirect, whether it occurs under physiological conditions, or whether it depends on additional binding partners. While the STRING database focuses on heterodimeric interactions, many proteins also function as homodimers or higher-order oligomers. A recent study explored the potential

for homo-oligomer formation at the proteome scale using AlphaFold2 Multimer predictions [45]. Extending our analysis to include such self-associations would be another valuable future direction.

Our results also underscore the pivotal role of intrinsically disordered regions in mediating PPIs at a genomic scale. Our findings reveal that over 98% of predicted high confidence PPIs involve at least some level of intrinsic disorder, from short IDRs (5–30 residues, 3307/3621) to long IDRs (over 30 residues, 2719/3621). This highlights IDRs as important players in cellular interactomes, potentially enabling dynamic and context-dependent binding [22,46,47]. For instance, the Ku80/Irbp complex simultaneously employs coil-to-order and conditionally folded binding, illustrating how structural versatility fine-tunes interactions in processes like the DNA damage response.

Evolutionarily, IDR interfaces exhibit remarkable diversity, from being highly conserved to lineage specific. However, the reliance on computational predictions warrants caution about misidentification and underidentification. While computationally predicted models often align well with experimental structures, such as the case of Ku80/Irbp in this study and the results from recent studies [18,28,48], these predictions still require careful interpretation, especially for IDRs due to their inherently dynamic and context-dependent nature [49]. When a predicted interaction involves an IDR, we advise caution, as such interactions might represent transient or context-dependent binding rather than stable interactions. In this sense, combining metrics such as pLDDT and PAE can provide useful indicators of local flexibility and uncertainty. On the other hand, by designing sophisticated, dedicated metrics, we may be able to distinguish true interactions at high precision [18,26,35,43,44,50–52]. Future development of specialized machine learning models for predicting the structure and function of IDRs [53,54], together with advances in high-quality experimental techniques [55], will enhance our ability to characterize the full spectrum of cellular PPIs – including both stable and transient interactions.

Note that in this study, we used cluster-based representative proteins from the STRING database. This can lead to discrepancies when a non-representative isoform is annotated as the interacting partner. Additionally, interacting partners form complexes that exceed the sequence-length limits of the computational resources available, preventing us from predicting the interactions between those pairs. These constraints may result in a reduced number of predicted interactions for a small subset of specific proteins. While these instances do not affect the overall conclusions of our study, they highlight an important limitation inherent to both the available functional annotations and current computational capabilities. Future work incorporating larger computational resources and improved functional annotations may help address these gaps.

## Methods

### Extracting representative PPIs from the STRING database

We first collected all PPIs with supporting evidence scores greater than 900 from the STRING database, version 11.5 [14]. We downloaded and extracted 107,946 PPIs with the highest STRING confidence scores (≥0.9). We used MMseqs2 [56] to cluster the proteins involved in these PPIs at 50% sequence identity and extracted PPIs from the representative proteins to further remove redundant protein sequences with high sequence similarities. This reduced the number of PPIs to 34,940 for structural predictions. Functional enrichment analysis was performed in the STRING database.

### Infer disordered, coil, and conditionally folded regions

We obtained the monomer structures of *D. melanogaster* proteome from the AlphaFold structural database, last accessed March 14, 2024 [57]. We used AlphaFold-disorder [41] to compute the intrinsic structural disorder (ISD) of protein monomers. In the monomers, we defines a region as disordered if five or more consecutive residues were predicted with ISD $> 0.5$. Regions with short, structured segments (<3 consecutive structured residues) embedded within disordered stretches were also considered disordered. Disordered regions with 30 or more residues were considered as long intrinsically disordered regions (long IDRs) [58], and those between 5 and 30 residues were short disordered regions (short IDRs).

We used DSSP [59] version 4.4 to obtain the secondary structures of the protein monomers. We further grouped the eight-code secondary structures defined in DSSP into three states [60] by converting H (α-helix), G ($3_{10}$-helix), and I (π-helix) to H (helix), E (β-strand) and B (bridge) to E (strand), and others to C (coil). We also extracted coil regions from the monomer structures where consecutive 5 or more residues were predicted as coils (C) in AlphaFold structural database. Similarly, regions with short helix or strand segments (fewer than 3 residues) embedded within coil stretches were also considered part of the coil regions. We further define conditionally folded (CF) regions as the regions that are predicted to have both high pLDDT (>70) and high structural disorder (ISD > 0.5) in their monomer states [41].

## Structure prediction and structural analysis of *D. melanogaster* PPIs

We used the full version of AlphaFold2 Multimer (version 2.3.0, with structural database identifiers provided in S5 Appendix) to predict the structures of the 34,940 *D. melanogaster* PPIs. For each prediction, we generated five models using different random seeds, and the best-scoring prediction was selected as the final model. In total, we were able to obtain 27,711 PPI predictions. The remaining pairs could not be predicted due to sequence-length constraints or memory limitations, but these failures were not associated with specific proteins or interaction types (S6 Fig). To obtain the interaction interfaces of each predicted PPI, we defined two residues – each from one protein in the predicted PPI structure – to be interacting if any pair of their non-hydrogen atoms were within 7 Å of each other. We excluded noisy, suspicious interactions when the predicted aligned error (PAE) between the two residues was greater than 15 Å, as large PAE values indicate uncertainties in their relative orientations. We further grouped the interacting residues into different interface regions by merging neighboring residues. Interface regions that are separated by fewer than 3 residues were considered contiguous and merged.

We again used DSSP [59] version 4.4 to obtain the secondary structures in the predicted PPI structures as described above. To detect coil-to-order transitions at coil interfaces, we compared the secondary-structure content of each interface in the monomer versus complex state using one-sided Fisher's exact tests. We defined coil-to-order binding if the content of secondary structures (helix [H] and strand [E]) increases significantly at a p-value cutoff of 0.05. Interfaces that reside or overlap with conditional folded (CF) regions were classified as conditional folded binding, or CF binding.

## Network analysis and visualization

We constructed, analyzed, and visualized the PPI network using the Python package Networkx [61]. We performed Markov Clustering (MCL) analysis [36] using a Python version of MCL, accessible at https://github.com/GuyAllard/markov_clustering, commit 28787cf. In our interactive web platform (https://melppi.github.io), we used Cytoscape API [62] to show the position of each protein pair in the predicted high confidence PPI network (pDockQ ≥ 0.5). The interactive PAE map and interface binding modes were visualized by Plotly [63].

## Sequence conservation analysis

We compiled a list of 15 genomes from *D. melanogaster* to *H. salinarum* (S1 Table). We used reciprocal best hits from blastp [64] to determine the orthologs of *Drosophila* proteins. We used the MAFFT-LINSI [65] version 7.429 to align the orthologous sequences. Sequence alignments were visualized by Mview [66].

## Structural comparison and visualization

We used FoldSeek-Multimer [67] with default parameters to search for similar PPI structures in the Protein Data Bank [68]. We then used US-align [69] to re-evaluate the TM-score values between the predicted complexes and the corresponding similar complexes identified by FoldSeek-Multimer. All structures were visualized by UCSF Chimera [70] and PyMol [71].

## Supporting information

**S1 Fig. Number of predicted PPIs in different STRING database categories.**
(TIF)

**S2 Fig. Overlap between high-confidence AlphaFold2 Multimer predictions, intermediate-confidence AlphaFold2 Multimer predictions, and the STRING database physical interactions. a).** The overlap between AlphaFold predictions and the highest-confidence STRING physical interactions (STRING physical confidence score ≥ 0.9). b). The overlap between AlphaFold predictions and the high-confidence STRING physical interactions (STRING physical confidence score ≥ 0.9).
(TIF)

**S3 Fig. High-confidence PPI network after removing ribosomal proteins.** (a) The modified network, obtained after removing 154 ribosomal proteins, including both cytoplasmic and mitochondrial, retains a similar overall layout to the original network (Figure 1b in the main text). (b) The modified network also shows a comparable connectivity distribution to that of the original. The connectivity distribution represents the number of proteins that have a given number of interaction partners. Note that the Y axis is in log scale, log2(Number of proteins + 1).
(TIF)

**S4 Fig. Top 10 MCL clusters of predicted high confidence PPIs.** Each cluster corresponds to an essential protein complex or a crucial cellular pathway.
(TIF)

**S5 Fig. Prediction details of the Ptp61F–Stat92E interaction.** (a) Left: full-length predicted model. Right: truncated model with the C-terminus of Ptp61F (residues 380–548), the N-terminus of Stat92E (residues 1–187), and the C-terminus of Stat92E (residues 696–961) removed for clearer visualization. (b) Model confidence (pLDDT) for the truncated model shown in (a). The dashed-line box highlights a region where binding involves a low-confidence segment of Ptp61F (residues 360–370). (c) PAE matrix. The blue off-diagonal region highlighted by the dashed-line box corresponds to the potential binding region indicated in (b).
(TIF)

**S6 Fig. A total of 34,940 predictions was attempted, of which 27,711 were successfully predicted and 7,229 failed due to memory limitations.** To determine whether prediction failures were randomly distributed across proteins, we performed a permutation test in which success labels were randomly reassigned 5,000 times while maintaining the total number of successful predictions. For each protein, empirical p-values were computed from the permutation distribution of success fractions and corrected for multiple testing using the Benjamini-Hochberg false discovery rate (FDR) method. The resulting FDR distribution shows that most proteins have FDR values close to 1, with only a small number of outliers, indicating that prediction failures were largely random rather than systematically associated with particular proteins or interaction types.
(TIF)

**S1 Table. Functional enrichment analysis of the largest connected subnetwork of predicted high-confidence PPIs.** Only GO terms with a false discovery rate smaller than 1e-50 were included.
(XLSX)

**S2 Table. Functional enrichment analysis of the second largest connected subnetwork of predicted high-confidence PPIs.**
(XLSX)

**S3 Table. Functional enrichment analysis of the third largest connected subnetwork of predicted high-confidence PPIs.** Only biological processes with a false discovery rate smaller than 1e-10 were displayed.
(XLSX)

**S4 Table. List of the 15 genomes to access the origin of IDR regions.**
(XLSX)

**S1 Appendix. Details of all 27,711 AlphaFold2 Multimer predictions, including ipTM, PAE, pDockQ, iLIS, and interface residues.**
(CSV)

**S2 Appendix. Interconnected subnetworks of the high-confidence PPI interaction network predicted by Alpha-Fold2 Multimer.**
(RTF)

**S3 Appendix. MCL clusters of the high-confidence PPI interaction network predicted by AlphaFold2 Multimer.**
(TXT)

**S4 Appendix. Interconnected subnetworks composed of protein pairs with limited or no STRING physical-network support (confidence score < 0.4) within the high-confidence AlphaFold2 Multimer predicted PPI network.**
(TXT)

**S5 Appendix. AlphaFold2 Multimer version 2.3 structural database identifiers.** This dataset contains 192,586 mmcif entries.
(TXT)

## Acknowledgments

We thank members of Zhao lab for helpful discussions. We thank The Rockefeller University High Performance Computing (HPC) Center for the support in computation.

## Author contributions

**Conceptualization:** Junhui Peng, Li Zhao.

**Data curation:** Junhui Peng.

**Formal analysis:** Junhui Peng.

**Funding acquisition:** Junhui Peng, Li Zhao.

**Investigation:** Junhui Peng.

**Resources:** Li Zhao.

**Visualization:** Junhui Peng.

**Writing – original draft:** Junhui Peng, Li Zhao.

**Writing – review & editing:** Junhui Peng, Li Zhao.

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
