## [Decision Letter · Decision Letter 0]

21 Oct 2025

A predicted structural interactome reveals binding interference from intrinsically disordered regions

PLOS Computational Biology

Dear Dr. Zhao,

Thank you for submitting your manuscript to PLOS Computational Biology. After careful consideration, we feel that it has merit but does not fully meet PLOS Computational Biology's publication criteria as it currently stands. Therefore, we invite you to submit a revised version of the manuscript that addresses the points raised during the review process.

Please submit your revised manuscript within 60 days Dec 21 2025 11:59PM. If you will need more time than this to complete your revisions, please reply to this message or contact the journal office at ploscompbiol@plos.org. Please include the following items when submitting your revised manuscript:

We look forward to receiving your revised manuscript.

Kind regards,

Ozlem Keskin

Academic Editor

PLOS Computational Biology

Shaun Mahony

Section Editor

PLOS Computational Biology

**Journal Requirements:**

4) We have noticed that you have referrred to Figure S2 in your manuscript. However, there is no corresponding file uploaded to the submission. Please upload it with the item type 'Supporting Information'.

5) We have noticed that you have uploaded Supporting Information files, but you have not included a list of legends. Please add a full list of legends for your Supporting Information files after the references list.

Potential Copyright Issues:

i) The following Figure contains a logo or branding: 5. We are not permitted to publish this under our CC-BY 4.0 license, even with permission. We ask that you please remove or replace it.

ii) The following Figure contains screenshots: 5. We are not permitted to publish these under our CC-BY 4.0 license, websites are usually intellectual property and are copyrighted.This includes peripheral graphics of the web browser such as icons and button. We ask that you please remove or replace it.

7) Thank you for stating that "All predicted structures are available" in "the DmelPPI website (https://jhpanda.github.io/DmelPPI_web/), which will be made public upon publication/acceptance." Please note that, though access restrictions are acceptable now, your entire data will need to be made freely accessible if your manuscript is accepted for publication. This policy applies to all data except where public deposition would breach compliance with the protocol approved by your research ethics board.

8) Please amend your detailed Financial Disclosure statement. This is published with the article. It must therefore be completed in full sentences and contain the exact wording you wish to be published.

9) Please provide a completed 'Competing Interests' statement, including any COIs declared by your co-authors. If you have no competing interests to declare, please state "The authors have declared that no competing interests exist". Otherwise please declare all competing interests beginning with the statement "I have read the journal's policy and the authors of this manuscript have the following competing interests:"

10) Please ensure that the funders and grant numbers match between the Financial Disclosure field and the Funding Information tab in your submission form. Note that the funders must be provided in the same order in both places as well. Currently, "Allen Distinguished Investigator Award from Paul G. Allen Family Foundation and the Shapiro-Silverberg Fund for the Advancement of Translational Research" are missing from the Funding Information tab.

**Reviewers' comments:**

Reviewer's Responses to Questions

**Comments to the Authors:**

**Please note that one review is uploaded as an attachment.**

Reviewer #1: uploaded as attachment

Reviewer #2: Review of PCOMPBIOL-D-25-01380

General comments

This is a carefully conducted study, which integrates structural information with data on the protein-protein interactions (PPI) network of D. melanogaster. An original aspect of this work is the investigation of the role of regions predicted to be disordered in the structures of the individual subunits and the detailed analysis of the contribution of these regions to the binding interfaces in the complex. The study deftly uses available DL-based tools (AlphaFold, AlphaFold2 multimer) to predict the structures of individual subunits and their pairwise hetero complexes; the limits of disordered regions in these structures is defined based on these tools. Data on PPIs is obtained from various well-established sources, and more particularly from STRING, which include functional associations, derived from data on co-expression and genomic context, in addition to those detected by experimental methods. Care is taken to concentrate on predicted structures with high confidence, to reduce noise in the analysis.

Results reveal important gaps between the detailed structure-based predictions and PPIs based on the evidence compiled in STRING. Notably of the ~28000 protein pairs with high confidence in STRING (≥0.9), only about a third (~29%) were predicted to form physical interaction at an acceptable confidence level based on the structural analysis. Of these, a mere 13% (~3600 pairs) were found to do so at a high confidence level. But reassuringly, the latter were found to be well supported by a reasonable physical interaction evidence in STRING. In addition, it was observed that many of the remaining 19% high confidence predicted complexes are labelled as functional associations in STRING, suggesting that a subset of such association may actually involve physically interacting protein pairs, as it has been suggested, and occasionally confirmed, for proteins in metabolic networks.

Interesting finding are also reported based on the detailed structural analysis of the contribution of IDR regions to shaping the interfaces of predicted complexes. Namely: that most of the interfaces in high confidence PPI comprises ordered as well as disordered or conditionally folded regions. This picture is complemented by a sequence conservation analysis of 15 genomes from different taxa and highlighting examples where IDR containing interfaces are conserved despite differences in disorder propensity and binding mechanism.

In conclusion, this is an interesting paper that provides new insights into the prevalent contribution of intrinsic disorder to the formation of protein complexes. It also demonstrates the value of detailed structural analyses in shaping the landscape of physically interacting proteins.

In the following are some specific comments:

1-The introduction merely lists citations of some large scale – proteome wide- studies on the role of intrinsic disorder in protein-protein interactions but lacks a succinct description of such studies. Including such description should help readers better understand the contribution of the present work.

2-Proteins frequently form homodimers, larger homo-oligomers, or quasi homo-oligomers (with paralogues). Such complexes may form different interfaces, including with other proteins. This situation was ignored here by investigating only hetero pairs formed by representative proteins! Ignoring information on paralogues is understandable since it is rarely if ever available from experimental methods or from the PPI DBs. On the other hand, the authors should check which proteins are likely to form homo-oligomers by using data from the study of Sweke et.al. (Cell 2024), which used AlphaFold to compute the atlas of homo-oligomers across the domains of life, and see what the impact might be on the results. Also, since many homo- dimers/oligomers tend to form co-translationally, it may be of interest to investigate the role of disorder in these complexes, to find out if it differs from the one in the heterodimers.

3-The described network analysis identifies several highly connected regions, which the authors report as containing many ribosomal proteins in addition to others. This is not surprising, as ribosomal proteins are often reported to ‘contaminate’ detected PPI and are therefore called ‘frequent flyers’. Due to their role in the translation machinery, they engage in many interactions, a significant fraction of which may be spurious (not evolved for function). The authors may want to delete these proteins, re-analyze the resulting network, and report the results if those differ.

4-AlphaFold-Disorder is only fleetingly mentioned in Ref 30 which is a review of databases of protein disorder. It probably involves regions of the polypeptide predicted with low confidence score by AlphaFold2-multimer. The exact procedure used should be succinctly outlined.

5- In line 235, it is stated that coil-to-order transition is more likely to involve induced fit. In light of the recent understanding of Allostery, which essentially rejects the induced fit as a valid mechanism, the coil-to-order transition may also involve conformational selection whereby, a very lowly populated structured conformer be selected, provided the interaction with the binding partner stabilizes it sufficiently.

6- I used the link provided in the text hoping to examine the interactive web interface, but unfortunately, except for the master Table on the Home page, none of the ‘view details’

links worked. The authors should make sure their interface is operational, including on a Mac, before resubmitting the revised version of their paper.

Reviewer #3: In this work entitled “A predicted structural interactome reveals binding interference from intrinsically disordered regions”, the authors present a large-scale prediction and systematic analysis of the Drosophila melanogaster interactome using AlphaFold2 Multimer combined with STRING functional and physical interaction datasets. Importantly, the authors provide an interactive web resource (DmelPPI).

Strength of the study is the integration of functional interactions. By not limiting the analysis to strictly physical interactions, the authors capture a broader landscape of potentially meaningful connections, many of which are supported by indirect evidence such as co-expression or text-mining. This inclusive approach, coupled with structural predictions, yields a resource that is valuable for both computational exploration and the generation of testable experimental hypotheses. The fact that many predicted complexes align well with known homologous structures adds confidence to the robustness of the pipeline. pDockQ has been used to asses AF2 prediction. Moreover, the authors place particular emphasis on the role of intrinsically disordered regions (IDRs), which they show are involved in the majority of high-confidence protein–protein interactions, often through coil-to-order transitions or conditional folding. Notably, the discrepancies obtained through the presented protocol have been carefully discussed and assessed.

The manuscript is overall well written, scientifically rigorous, and of clear relevance to the computational biology community. It offers a good contribution by not only cataloging predicted interactions but also highlighting mechanistic roles of IDRs, a feature that is often overlooked in large-scale structural interactome studies. The interactive website further enhances its value as a community resource.

I consider the work suitable for publication in PLOS Computational Biology.

Minor comments:

• As stated by the authors “While computationally predicted models often align well with experimental structures … these predictions still require careful interpretation, especially for IDRs due to their inherently dynamic and context-dependent nature.” In this context, I think it would be of usefull if the authors provide guidelines/opinions on how the think the interpretation should be lead (taking into account the variability of single cases).

• It may be useful to briefly discuss why AF2 and not AF3 has been used.

• On page 6 line 38, the notation “etc.” does not give accuracy to the whole sentence. I suggest to extended the given examples.

**Have the authors made all data and (if applicable) computational code underlying the findings in their manuscript fully available?**

Reviewer #1: **No:** repository should include all necessary files (or at least Zenodo links), such as the input sequences, the predicted models, and the scripts used to process the data and generate the figures clearly and logically. I think GitHub would benefit a lot from much clearer instructions. From the current standpoint, I would be unable to reproduce the results without putting major work into them and writing a lot on my own

Reviewer #2: Yes

Reviewer #3: Yes

PLOS authors have the option to publish the peer review history of their article (what does this mean? ). If published, this will include your full peer review and any attached files.

**Do you want your identity to be public for this peer review?** For information about this choice, including consent withdrawal, please see our Privacy Policy .

Reviewer #1: No

Reviewer #2: No

Reviewer #3: No

**Figure resubmission:**
---

## [Decision Letter · Decision Letter 1]

8 Jan 2026

Dear Dr. Zhao,

We are pleased to inform you that your manuscript 'A predicted structural interactome reveals binding interference from intrinsically disordered regions' has been provisionally accepted for publication in PLOS Computational Biology.

Best regards,

Ozlem Keskin

Academic Editor

PLOS Computational Biology

Shaun Mahony

Section Editor

PLOS Computational Biology

Reviewer's Responses to Questions

**Comments to the Authors:**

Reviewer #2: The revised version has adequately addressed my comments and suggestions and can be accepted for publication.

Reviewer #3: Here, the authors present a deeply and extensively revised version of the work entitled “A predicted structural interactome reveals binding interference from intrinsically disordered regions”. It is really commendable the amount of work behind this revised version of the paper. The present version of the paper is substantially clearer in every aspect compared to the original version and the authors addressed all the relevant concerns. In this revised version, all the highlighted unclear sections have been rewritten improving the overall accessibility of the presented data.

Given all of the above, I consider the present version of the work suitable for publication in Plos Computational Biology without any change.

**Have the authors made all data and (if applicable) computational code underlying the findings in their manuscript fully available?**

Reviewer #2: Yes

Reviewer #3: Yes

PLOS authors have the option to publish the peer review history of their article (what does this mean? ). If published, this will include your full peer review and any attached files.

**Do you want your identity to be public for this peer review?** For information about this choice, including consent withdrawal, please see our Privacy Policy .

Reviewer #2: **Yes:** Shoshana Wodak

Reviewer #3: No

---

## [Editor Report · Acceptance letter]

PCOMPBIOL-D-25-01380R1

A predicted structural interactome reveals binding interference from intrinsically disordered regions

Dear Dr Zhao,

I am pleased to inform you that your manuscript has been formally accepted for publication in PLOS Computational Biology. Your manuscript is now with our production department and you will be notified of the publication date in due course.

With kind regards,

Zsofia Freund
